# Imaging or Adrenal Vein Sampling Approach in Primary Aldosteronism? A Patient-Based Approach

Irene Tizianel [1,2], Chiara Sabbadin [2], Caterina Mian [1,2], Carla Scaroni [1,2] and Filippo Ceccato [1,2,*]

1   Department of Medicine DIMED, University of Padova, 35128 Padova, Italy
2   Endocrine Disease Unit, University-Hospital of Padova, 35128 Padova, Italy
*   Correspondence: filippo.ceccato@unipd.it

**Abstract:** Primary aldosteronism (PA) is the most frequent cause of secondary hypertension, associated with an increased risk of cardiovascular and cerebral disease, compared to essential hypertension. Therefore, it is mandatory to promptly recognize the disease and offer to the patient the correct diagnostic–therapeutic process in order to reduce new-onset cardiovascular events. It is fundamental to define subtype classification (unilateral or bilateral disease), in order to provide the best treatment (surgery for unilateral and medical treatment for bilateral disease). Here, we report five clinical cases of different subtypes of PA (patients with monolateral or bilateral PA, nondiagnostic AVS, allergy to iodinated contrast, and patients not suitable for surgery), with particular attention to the diagnostic–therapeutic process and the different approaches tailored to a single case. Since PA is a spectrum of various diseases, it needs a personalized diagnostic–therapeutic process, customized for the individual patient, depending on previous medical history, suitability for the surgery and patient's preferences.

**Keywords:** primary aldosteronism; surgery; adrenal vein sampling



## 1. Introduction

Primary aldosteronism (PA) is the most common endocrine form of secondary hypertension, and it is associated with enhanced risk of cardiovascular events [1,2]. Its prevalence is up to 10% of hypertensive patients, further increased in high-risk populations (up to 20% of subjects with resistant hypertension, or 5% of hypertensive patients with sleep apnea syndrome [2–4]).

PA is a group of disorders characterized by an excess of aldosterone secretion; it is identified by inappropriate mineralocorticoid activity, independent from sodium and volemic status, not controlled by the major regulators (plasma potassium levels and renin–angiotensin system). The aldosterone excess leads to hypertension, sodium retention, hypokalemia, atrial fibrillation, and cardiovascular damage [1,5]. It is noteworthy that a consistent part of cardiovascular damage in PA depends not only on the effects of arterial blood pressure (BP), but also on cardiac abnormalities, like left-ventricular hypertrophy and myocardial fibrosis, which are caused by elevated aldosterone levels [6].

According to international Endocrine Society (ES) Clinical Practice Guidelines, case detection of PA is recommended in hypertensive patients aged <45 years, in case of resistant hypertension (defined with the use of at least three conventional antihypertensive drugs, including a diuretic), hypertension and spontaneous or diuretic-induced hypokalemia, hypertension and adrenal incidentaloma, hypertension and sleep apnea, hypertension and a family history of early-onset hypertension or cerebrovascular accident at a young age (<40 years), and all hypertensive first-degree relatives of patients with PA [1].

The diagnostic–therapeutic process in PA constitutes various steps: first screening tests in high-risk patients, then identification of a PA subtype, and finally choice of treatment (adrenalectomy in case of monolateral secretion; otherwise, medical therapy with

mineralocorticoid receptor antagonists, MRAs) [4]. Therefore, subtype classification is fundamental, in order to suggest the best treatment. Nonetheless, the two subtypes of PA (unilateral and bilateral) represent the two extremes of the entire spectrum of the disease: true unilateral disease without any contralateral aldosterone secretion is rare (also in case of low/suppressed renin levels), while different lateralization in bilateral PA is a common condition [4,7]. Targeted treatments with unilateral adrenalectomy or medical therapy reduce the risk of adverse cardiovascular events and can provide a reversal of the organ damage, especially if renin levels rise to normal [1,8].

Regarding surgical approach, minimally invasive laparoscopic adrenalectomy is the treatment of choice in patients with PA, because it is safe and effective in the resolution of arterial hypertension and hypokalemia correction, with a lower rate of postoperative complications compared to open adrenalectomy (in experienced hands). Additional benefits of laparoscopic adrenalectomy are smaller incisions, lower postoperative pain, and shorter hospitalization [9]. Therefore, since the treatment of choice of monolateral forms of PA is unilateral adrenalectomy, in a setting of "*precision medicine*", all efforts should be currently addressed to select the correct adrenal to remove. Indeed, some cases of non-secreting adrenal adenomas not responsible for aldosterone secretion, in patients with unilateral PA, are described. During routine clinical practice, in the diagnostic process of PA, the "four corners criteria" should be fulfilled when considering the diagnosis of unilateral forms: (1) the biochemical demonstration of PA, (2) the lateralization of aldosterone secretion either at AVS or I-131 adrenal scintigraphy, (3) the demonstration of adenoma at pathology assessment, and (4) the resolution of hypokalemia and cure/improvement of arterial hypertension during follow-up after adrenalectomy [10,11]. It is important to remember point 4 because it defines remission and confirms the initial diagnosis.

In this narrative review, we present the subtype diagnosis of PA in the setting of a clinical practice patient-based approach. The keywords used for the MEDLINE/PubMed research (in July 2022, with MeSH tree terms) were the following: primary aldosteronism, adrenal vein sampling, adrenalectomy, and mineralocorticoid receptor antagonist. The search was performed by two authors (F.C. and I.T.) independently; additional relevant articles were identified using manual searches and included a thorough review of other meta-analyses, review articles, and relevant references. Our aim is to describe the diagnostic tools that can be used, alone or combined, before the treatment's choice: surgery for unilateral and medical treatment for bilateral forms, according to the patient's clinical history.

## 2. Adrenal Vein Sampling and Unconventional Adrenal Indices

Adrenal vein sampling (AVS) is the gold standard for the diagnosis of subtypes in PA; it is recommended by all international guidelines and consensus statements. In our opinion, AVS should be considered in patients with a high likelihood of post-surgical remission, such as those PA patients who have a high probability of unilateral disease and are willing to have a potential surgical cure, achieving a long-life remission [1,12].

AVS is not widely available because it is a challenging procedure, is poorly standardized, and must be performed by an interventional radiologist with sufficient expertise. It is important to keep in mind that cannulation of the right adrenal vein is cumbersome because of the anatomical origin, diameter, presence of variants, and direct integration into the inferior vena cava (IVC). However, in experienced hands, the technical success rate of AVS can be close to 90%. It is made using a percutaneous femoral approach and adrenal veins are catheterized simultaneously or sequentially [7]. Blood is obtained from both the adrenal veins and from the IVC, and it is assayed for aldosterone and cortisol concentrations. The cortisol concentrations from the adrenal veins and IVC are usually used to confirm successful catheterization. The adrenal vein-to-IVC cortisol ratio should be at least 2:1. Dividing the right and left adrenal vein aldosterone concentrations by their respective cortisol concentration provides cortisol-corrected ratios. Then, dividing the cortisol-corrected aldosterone ratio from the right and left vein provides the lateralization index. A lateralization index >4 indicates unilateral aldosterone excess, whereas a

lateralization index <3 may suggest bilateral aldosterone excess; values between 3 and 4 are indeterminate [7,13]. According to an international study, 46% of centers use synthetic ACTH (cosyntropin; ACTH 1–24) infusion during AVS. The rationale to perform an ACTH-stimulated AVS is to minimize stress-induced fluctuations in aldosterone secretion during non-simultaneous AVS, to maximize the cortisol gradient from adrenal vein to IVC and, thus, confirm correct sampling, and to maximize the normal adrenal secretion of aldosterone from aldosterone-producing adenomas [14].

Complications of AVS are rare and include groin hematoma, adrenal hemorrhage, and adrenal vein dissection [15]. Although AVS can be performed regardless of concomitant medications if plasma renin activity remains suppressed [16], we suggest the washout of interfering drugs. The aldosterone-producing cell cluster (APCC) is one of the main pitfalls of AVS. Briefly, APCC is a group of cells positive for focal *CYP11B2* expression (aldosterone synthase gene) in the subcapsular portion of the human adult adrenal cortex. It produces and secretes excessive amounts of aldosterone, but is so small that even modern imaging techniques are unable to localize it [17].

The lateralization index has been validated and can be correctly interpreted only in cases of bilaterally selective AVS. Therefore, in the last few years, unconventional indices have been studied to define the lateralization of aldosterone secretion even in the case of incorrect cannulation of one (especially the right) or both adrenal veins. A recent multicenter Italian study aimed to evaluate the reliability of two unconventional indices in predicting lateralization during unstimulated AVS. These unconventional indices are the monolateral index (MI), defined as the aldosterone/cortisol ratio in the adrenal vein divided by the aldosterone/cortisol ratio in IVC, and the monoadrenal index (MAI), defined as the aldosterone/cortisol ratio in the adrenal vein. The diagnostic accuracy of predicting unilateral disease of MAI and MI was performed by classification of each adrenal into three different conditions (ipsilateral, contralateral, or bilateral hypersecretion). In conclusion, they demonstrated the high diagnostic accuracy of the unconventional indices (lesion and hypokalemia corrected) in predicting unilateral disease in a large cohort of PA patients subtyped with AVS [18].

## 3. Conventional and Nuclear Imaging

Although AVS remains the gold standard for the diagnosis of subtype in PA, alternative strategies for the demonstration of lateralization are employed in clinical practice, including conventional imaging and nuclear medicine techniques.

The first step for PA subtyping is adrenal CT, which can show normal adrenal glands, unilateral adenoma, bilateral adenoma, bilateral micro- or macronodular adrenal hyperplasia, or adrenal carcinoma (in rare cases). The role of CT is useful in clinical practice because, in a young person with an evident diagnosis of PA and unilateral adenoma, AVS can be bypassed, as well as in bilateral adrenal hyperplasia due to the high suspicion of bilateral PA. Adrenal incidentalomas are uncommon in young people; the finding of a unilateral adrenal adenoma, usually <2 cm, in a young patient with PA (<35 years), is suggestive of unilateral disease [19]. Contrariwise, considering the high prevalence of adrenal incidentalomas in adults and in the elderly, the accuracy of CT in the diagnosis of PA is low [4].

Lipid content measurement with Hounsfield Units (HUs) and contrast washout are not able to provide sufficient information about the secretory nature of the adrenal nodules [20,21]; MRI is a second-choice technique due to its reduced spatial resolution [13]. Historically, I-iodomethyl-norcholesterol (NP-59) was used as a scintigraphic tracer for the detection of 3onoliteral forms of PA. In order to suppress the tracer uptake by the nonautonomous adrenal cortex, it is necessary to administer high-dose dexamethasone for a prolonged period before the exam (usually 8 mg per day for at least 7 days). Therefore, glucocorticoid suppression, sequential imaging (one scan daily for 4–7 days), reduced resolution for small adenomas, and low availability of the tracers limit the use of NP-59 scintigraphy, and it is actually not considered an alternative to AVS in most centers [22].

$^{11}$C-metomidate PET is an interesting option for functional imaging. It uses $^{11}$C-metomidate, a strong inhibitor of CYP11B1 (11-beta-hydroxylase) and CYP11B2 (aldosterone synthase) activity in the adrenal cortex [23]. In the past few years, several authors have studied the role of $^{11}$C-metomidate PET in the diagnosis of subtyping of PA as an alternative to AVS. Sonio et al. recently found that $^{11}$C-metomidate PET-CT had lower sensitivity and specificity to detect lateralization in comparison to AVS. Their analysis showed that the concordance between $^{11}$C-metomidate PET-CT and adrenal CT was low in both the lateralizing and the non-lateralizing groups [23]. On the contrary, Burton et al. and O'Shea et al. recently found that $^{11}$C-metomidate PET provided useful information to guide decision making. Burton et al. found that the SUV max ratio cutoff of 1:25 to 1 provides 87% specificity and 76% sensitivity in subtyping PA [24,25].

## 4. Imaging Versus AVS in the Diagnosis of Subtyping

Several studies have analyzed the different roles of imaging and AVS in the diagnosis of subtyping of PA. The SPARTACUS trial, a prospective randomized study comparing AVS with CT-based decisions on treatment, reported no significant differences between the postoperative need for antihypertensive drugs and the quality of life in 184 patients with PA [26]. Conversely, an international multicentric retrospective nonrandomized study found a biochemical remission ratio in 80% (188 of 235) cases after a CT-based treatment decision vs. 93% (491 of 526) after an AVS-based treatment decision. Clinical and biochemical remissions were assessed on postoperative BP measurement, antihypertensive drug dosage, aldosterone-to-renin ratio (ARR), and normalization of hypokalemia. Therefore, the authors concluded that imaging-based diagnosis leads to a low probability of achieving complete biochemical remission compared with AVS-based diagnosis [27]. Another recent multicenter international study on 1311 patients confirmed these data, reporting that imaging did not provide an accurate diagnostic value in PA, especially in unilateral PA. In this cohort, cross-sectional imaging was not able to identify a lateralized cause of disease in about 40% of PA patients and failed to identify the culprit adrenal in 28% of patients with unilateral PA [28].

## 5. Clinical Cases Presentation

Here, we report five different clinical cases of different subtypes of PA (monolateral, bilateral, nondiagnostic AVS, allergy to iodinated contrast, and not suitable for surgery) diagnosed and treated at our Endocrine Unit, with particular attention paid to the diagnostic–therapeutic process and the different approaches tailored on the single case, as reassumed in Figure 1.

### 5.1. Case 1

A 48 year old woman arrived at our center in May 2019 complaining about arterial hypertension (mean BP 160/100 mmHg) for 12 months, treated with olmesartan, and associated hypokalemia (2.9–3 mmol/L). With suspicion of PA, at the first clinical evaluation, she presented a CT scan with a left adrenal adenoma of 13 mm. Olmesartan therapy was switched to amlodipine and doxazosin, in order to perform ARR without any interference. The result was a pathological ARR: elevated aldosterone level (456 pmol/L) and suppressed renin. The patient underwent confirmatory tests; a saline infusion test (SIT) and captopril challenge test (CCT) confirmed the diagnosis of PA, and the aldosterone plasma levels after SIT were 948 pmol/L (we consider in our center a pathological case if >140 pmol/L), with suppression of aldosterone levels after CCT <30% (aldosterone levels in orthostatic position–561 pmol/L; aldosterone levels after captopril 50 mg–556 pmol/L, as shown in Table 1).

## Subtyping of primary aldosteronism

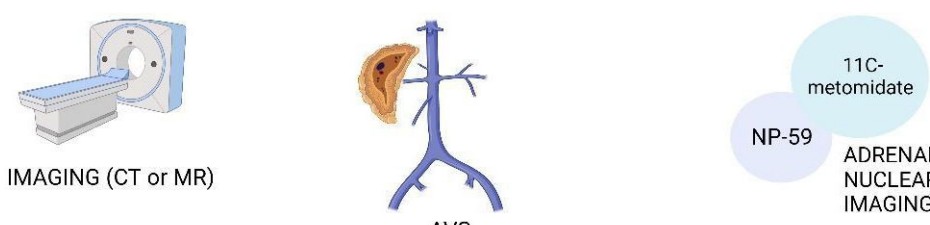

IMAGING (CT or MR)    AVS    NP-59    11C-metomidate    ADRENAL NUCLEAR IMAGING

## Clinical Questions

○ Unilateral or bilateral disease
○ Controindication to surgery
○ Age
○ Comorbidities
○ Hypertension duration

### SURGERY

• monolateral disease
• young age
• short hypertension duration

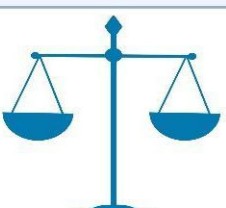

### MRAs

• bilateral disease
• patient not suitable or not willing to perform surgery
• comorbidities which prevent surgery

**Figure 1.** The balance of the diagnostic–therapeutic process in patients with PA. In our opinion, the choice between the two treatments (surgery or medical therapy with mineralocorticoid receptor antagonist, MRA) should be guided by clinical questions (summarized in the light blue box) matched with the process of the subtype of PA (using the imaging techniques depicted in the first part of the figure). Created with BioRender.com, accessed on 7 November 2022.

**Table 1.** Clinical cases description.

| Case | Confirmatory Test | Imaging | AVS | Treatment | Follow-Up |
|---|---|---|---|---|---|
| 1 | SIT and CCT: both positive | Left adrenal adenoma | Left-sided aldosterone lateralization | Left adrenalectomy | Good control of BP on Olmesartan therapy, renin levels no more suppressed 6 months after surgery |
| 2 | SIT and CCT: both positive | Left adrenal adenoma | AVS not interpretable for inadequate catheterization. Use of unconventional indices (MAI and MI) with demonstration of left aldosterone lateralization | Left adrenalectomy | Normotension and normal potassium levels at last follow-up (3 months) |
| 3 | SIT: positive | CT: left adrenal adenoma Adrenal scintigraphy: bilateral tracer uptake, larger on the left side | Left aldosterone lateralization (AVS performed with previous DEX premedication) | Left adrenalectomy | Normotension without any pharmacological treatment after surgery |
| 4 | CCT: positive | Right adrenal adenoma | AVS not performed because the patient was not suitable for surgery | Medical therapy with MRA (potassium canrenoate) | Good control of BP with MRA since last follow up (30 months); no worsening of the hypertensive cardiomyopathy |

**Table 1.** *Cont.*

| Case | Confirmatory Test | Imaging | AVS | Treatment | Follow-Up |
|------|-------------------|---------|-----|-----------|-----------|
| 5 | CCT: positive | Bilateral adrenal adenoma (20 mm right and 15 mm left) | Right aldosterone lateralization | Scheduled for right adrenalectomy | MRA therapy in association with Ca-antagonist with good pressure control |

Abbreviations: BP: blood pressure, DEX: dexamethasone, SIT: saline infusion test, CCT: captopril challenge test, MRA: mineralocorticoid receptor antagonist, AVS: adrenal venous sampling, MAI: monoadrenal index, MI: monolateral index.

To evaluate the surgical approach, she underwent AVS, which demonstrated a left-sided aldosterone lateralization, according to radiological findings, and 50 mg spironolactone therapy was initiated. Minimally invasive left adrenalectomy was performed in August 2020, without complications. After surgery, potassium serum levels were normal, and therapy with olmesartan was maintained, with good BP control. Renin levels appeared no more suppressed 6 months after surgery, confirming PA remission.

Does True Unilateral PA Exist?

Subtyping of PA should be performed as soon as possible, in the diagnostic work-up of PA, to define whether the patient is suitable for surgery; unilateral adrenalectomy is recommended in patients with a monolateral form of PA that agree with surgery. In recent years, the dichotomous division of PA into unilateral and bilateral forms has been questioned due to the demonstration of various adrenal histopathological phenotypes [29]. These nonclassical adrenal reports have been discovered through immunohistochemistry and, in recent studies, have been described in 5–25% of PA cases [30]. They are characterized by multiple aldosterone-producing micronodules (called APCCs) and aldosterone-producing diffuse hyperplasia [31–33]. Nonetheless, several studies recently demonstrated that patients with unilateral PA treated with adrenalectomy had a favorable outcome in terms of cardiovascular risk, compared to those treated with medical therapy [34,35]. These studies highlight that true unilateral forms of PA exist and benefit from surgical management [29]. In the authors' opinion, the discovery of a selective monolateral aldosterone excess after AVS should guide surgery, also applicable in the case of contralateral adrenal (nonfunctioning) adenoma. Regarding APCCs, which are held responsible for a spectrum of bilateral forms of PA, a recently published study reported the transcriptome of APCCs at single-cell resolution. The authors performed sc-RNA-seq of human adult adrenals; they deduced that APCCs develop from the zona glomerulosa and appear to produce aldosterone autonomously through the expression of CYP11B2. Moreover, somatic mutations are frequently harbored, which are also found in aldosterone-producing adenomas (APAs); thus, some APCCs can be the precursor of APAs [36].

*5.2. Case 2*

A 62 year old woman arrived at our center complaining about arterial hypertension from 2008, treated with losartan and lercanidipine, and hypokalemia from 2 years (K 3.1 mmol/L). A few years before, she underwent thorax CT for persistent cough, and an adrenal left adenoma of 14 mm was incidentally discovered. An accurate wash-out from angiotensin receptor blockers of 4 weeks was performed, together with correction of hypokalemia. The ARR resulted positive, and confirmatory tests (SIT and CCT) corroborated the diagnosis of PA; therefore, AVS was performed (resumed in Table 2).

**Table 2.** AVS results for Case 2.

|  | Interpretation | Baseline | 15 min |
|---|---|---|---|
| Ratio dx/sin | lateralization index > 3 | 0.01 | 0.01 |
| Ratio sin/dx | lateralization index > 3 | 83.36 | 78.31 |
| Cortisol right adrenal/cortisol VCI | selectivity index > 2 | 0.95 | 0.93 |
| Cortisol left adrenal/cortisol VCI | selectivity index > 2 | 3.82 | 5.05 |

Are These AVS Results Interpretable?

The conventional indices used for the assessment of aldosterone lateralization are the lateralization index, defined as the dominant vs. nondominant adrenal vein aldosterone/cortisol ratio, and the contralateral index, defined as the nondominant adrenal vein aldosterone/cortisol ratio. However, these indices are truly reliable and have been validated only in the case of bilaterally selective AVS (defined as a bilateral selectivity index >2) [1,18]. Therefore, in recent years, new unconventional indices have been proposed for the assessment of aldosterone validation even in the case of a lack of selectivity of AVS, due to the incorrect cannulation of one or both adrenal veins [18]. In our patient, as shown in Table 2, the selectivity index showed an incorrect catheterization of the right adrenal vein, altering the interpretation of the right lateralization index. Unconventional indices MAI and MI were calculated and depicted in Table 3; three out of four unconventional indices confirmed the left lateralization of aldosterone secretion, providing indication of a left adrenalectomy. In fact, except for right MI, the other three indices were concordant with the hypokalemia reference values of MAI and MI, which refer to potassium levels measured the day of AVS. Minimally invasive left adrenalectomy was performed in July 2022, without complications. After surgery serum potassium and BP levels were normal.

**Table 3.** Unconventional indices results Case 2.

|  | MAI (Monoadrenal Index) | MI (Monolateral Index) |
|---|---|---|
| left | 52.34 | 30.78 |
| right | 0.6 | 0.78 |

*5.3. Case 3*

A 50 year old woman arrived at the first evaluation in our Endocrine Unit in 2019 because of a previous diagnosis of PA due to resistant hypertension and hypokalemia. She had already performed SIT, which confirmed PA diagnosis, and abdominal CT, which demonstrated a small left adrenal adenoma (10 mm). In previous years, she did not tolerate therapy with spironolactone for menstrual spotting and had an allergic reaction to iodinated contrast media (ICM) during CT; therefore, she performed a subsequent iodocholesterol scintigraphy which demonstrated a bilateral tracer uptake, greater on the left side. During iodocholesterol scintigraphy, she had an anaphylactic shock. We decided to perform AVS after glucocorticoid therapy (Table 4). According to guidelines, to determine the aldosterone lateralization ratio, we should divide the higher adrenal vein cortisol-corrected aldosterone ratio by the lower adrenal vein cortisol-corrected aldosterone ratio. In this case, this was not feasible due to the administration of glucocorticoids prior to AVS; hence, during the procedure, we also measured plasmatic catecholamine, 4-androstenedione, and 17-alpha-hydroxyprogesterone. Data shown in Table 4 represent the aldosterone lateralization ratios with the use of 17-alpha-hydroxyprogesterone levels measured during AVS. Left PA was consistent with CT and scintigraphy findings; hence, the patient was referred to endocrine surgery, where histological examination confirmed an adrenal adenoma, achieving normalization of BP levels.

**Table 4.** AVS results for Case 3 (aldosterone lateralization measured with 17 hydroxyprogesterone levels).

|  | Interpretation | Baseline | 15 min |
|---|---|---|---|
| Ratio dx/sin | lateralization index > 3 | 0.01 | 1.16 |
| Ratio sin/dx | lateralization index > 3 | 109.6 | 3.26 |

Can AVS Be Feasible (Safe for Patients and Informative for Physicians) in Patients with Iodine Contrast Media Allergy?

Performing AVS in patients with ICM allergy is a challenge; therefore, alternative options have been proposed in the last few years: use of gadolinium-based contrast media for AVS; carbon dioxide ($CO_2$) as a substitute for contrast dye or use of adrenal scintigraphy with radioiodine-labeled cholesterol analogue. However, all these options have some limitations [37–39]. Oral premedication with glucocorticoids in patients with suspected or known ICM allergy is traditionally performed with prednisone (usually three doses of 50 mg at 13, 7, and 1 h before the procedure) in association or not with nonselective antihistaminic drugs [40]. In the case of AVS, prednisone could be neglected because it makes AVS results difficult to interpret, due to its interference with cortisol immunoassay and the suppression of basal cortisol levels [38]. Recently, Younes et al. reported their experience on seven patients with PA and a known allergy to ICM, who underwent successful AVS using dexamethasone premedication (which is not measured as cortisol in immunoassay). Nonetheless, all seven patients presented a normal post-ACTH cortisol response during stimulated AVS, despite suppressed basal peripheral cortisol levels; moreover, all seven patients had bilaterally selective AVS, based on basal and post-ACTH selectivity index [41]. Therefore, dexamethasone premedication for AVS in patients with known ICM allergy is safe and can achieve reliable AVS results for diagnosis of subtyping of PA.

There are also other methods of performing AVS. For example, in the case of adrenal Cushing syndrome due to bilateral adrenal masses, catecholamine concentrations in peripheral and adrenal veins are used for determining correct catheterization, as well as adrenal androgens [42]. Catecholamine measurement can be an alternative in patients with ICM allergy, especially those premedicated with dexamethasone [41]. Recently, Ceolotto et al. reported that the SI calculated with metanephrine or androstenedione was at least threefold higher than SI calculated with cortisol, increasing the rate of bilaterally successful AVS [43]. Moreover, metanephrine-based SI is better in determining the correct catheter tip location, as confirmed by injecting contrast media to show retrograde adrenal tissue [44].

*5.4. Case 4*

A 52 year old man arrived at our center in 2013 with suspicion of PA. He complained of two previous brain hemorrhages in 2012 and 2013, secondary to uncontrolled hypertension with hypokalemia. His pharmacological therapy was ramipril and hydrochlorothiazide; he was then switched to nifedipine and doxazosine to perform ARR and CCT, which resulted diagnostic for PA. Adrenal CT showed a right adrenal adenoma of 8 mm; cardiological evaluation showed hypertensive cardiomyopathy. Given the persistence of hypokalemia, treatment with potassium canrenoate was started. We inquired whether to perform AVS for diagnosis of subtyping and subsequent eventual unilateral adrenalectomy or to continue medical treatment with MRAs.

BP levels were adequately controlled with MRAs and nifedipine, right adrenal adenoma size and features were consistent with an adenoma [45], and serum potassium was normal. Nonetheless, the patient had previous cardiovascular complications and an elevated surgical risk; therefore, after multidisciplinary discussion, it was preferred to continue with medical therapy with MRAs, achieving adequate BP and potassium levels.

Are MRAs Better Than Surgery?

Although monolateral PA is considered a resectable disease, in many cases, hypertension can persist after treatment [6,46]. Treatment with MRAs is currently recommended for

bilateral forms of PA; however, prospective studies showed that MRAs have therapeutic value comparable to surgery in the long term. In a recent review of six studies which compared PA patients treated with MRAs or surgery, the effects on arterial blood pressure did not differ between the two treatments [47].

Normalization of arterial BP and hypokalemia resolution are not the only goals of PA treatment; prevention of subclinical organ damage leading to cardiovascular, renal, and metabolic complications is of utmost importance [48]. In such a scenario, a retrospective study reported data of 270 PA patients (unilateral or bilateral forms) treated with unilateral adrenalectomy or MRAs. After a median follow-up of 12 years, the incidence of cardiovascular events did not differ between the group treated with surgery and the group treated with medical therapy [49]. Moreover, MRAs were demonstrated to resolve subclinical organ damage and metabolic comorbidities, as well as reduce the risk of cardiovascular events and renal disease progression. To conclude, MRA treatment is a valid option of treatment in bilateral PA forms and patients not suitable or not willing to perform adrenalectomy [50].

### 5.5. Case 5

A 57 year old woman arrived at our center in 2022 with a probable diagnosis of PA, due to resistant hypertension from 2014 and associated hypokalemia from 2021. In 2019, after an accidental fall, she performed an abdominal CT which described bilateral adrenal adenomas. Her pharmacological therapy was MRAs and barnidipine, with good control of arterial hypertension. After an adequate wash-out from MRAs, ARR and CCT tests were performed resulting diagnostic for PA. An adrenal CT scan was repeated with the finding of stable bilateral adrenal adenomas (20 mm right and 15 mm left) with HU <10. A cardiological evaluation concluded the absence of cardiopathy. She underwent AVS which demonstrated a clear right lateralization of the aldosterone secretion. After this finding, in collaboration with our expert radiologist of the multidisciplinary adrenal team, we revised adrenal CT imaging and found that the right adrenal adenoma showed a great iodine contrast uptake in both arterial and venous phases, as shown in Figure 2. After the discharge, the patient maintained a combined MRA and calcium-antagonist therapy, achieving a good control of BP; she is scheduled for right adrenalectomy.

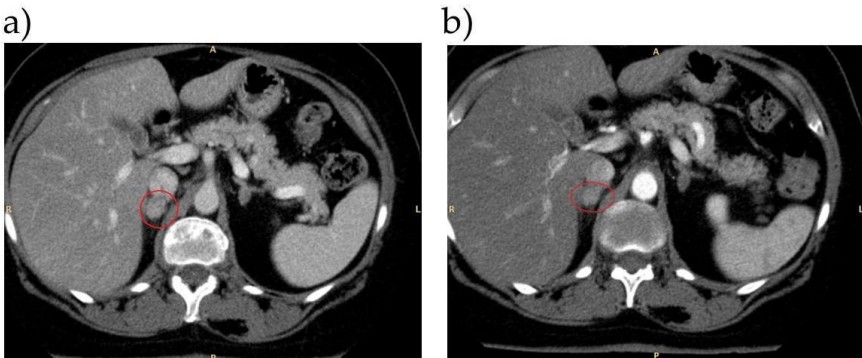

**Figure 2.** Arterial CT (panel (**a**)) and venous (panel (**b**)) phase.

### 6. Conclusions

PA is a frequent cause of secondary hypertension, associated with an increased risk of cardiovascular disease, compared to essential hypertension. Therefore, it is mandatory to promptly recognize the disease and offer to the patient the correct diagnostic–therapeutic process [4,51]. After a correct diagnosis of adrenal aldosterone excess, the treatment of PA requires a skillful combination of imaging and clinical practice, because there are several aspects to consider, as shown in Table 5.

According to international guidelines, patients with aldosterone lateralization during AVS should be addressed with unilateral adrenalectomy, because surgical treatment is curative and is more effective than medical therapy with MRAs (especially when renin levels remain suppressed [8,52]).

For the diagnosis of subtyping in PA, AVS is still the gold standard and should be performed by an expert radiologist in a high-volume center. Since adrenal incidentalomas are common findings in patients with cardiovascular events [53], all efforts should be considered to diagnose PA in these patients. Recently, the discovery of unconventional indices has overcome the impossibility of interpretation of AVS results in which there is incorrect cannulation of one or both adrenal veins. AVS is usually performed with ICM; dexamethasone premedication does not interfere with AVS result and makes AVS a safe procedure in patients with known or suspected allergy.

Our work presented some limitations. The design did not consider a systematic review; in the literature, there are only few evidence-based studies regarding the comparison of different approaches for PA subtyping or treatment. Nonetheless, a practice real-world application of the diagnosis of PA is needed, and we suggest a patient-based approach with some clinical cases.

In conclusion, PA is a spectrum of various diseases and needs a tailored diagnostic–therapeutic process, customized for the individual patient, depending on previous medical history, suitability for surgery, and patient's preferences. AVS is a safe and informative procedure, and it should be considered in all patients with PA who are good surgical candidates for 10onoliteral adrenalectomy.

**Table 5.** Key points of the review, presented in the five clinical cases presented, and most relevant studies in the literature.

| First Author | Cohort Described | Significant Findings |
|---|---|---|
| AVS is better than imaging to define the subtyping of PA | | |
| Williams TA [27] | 761 patients with unilateral PA (235 with CT management diagnosed from 1994–2016, and 526 with AVS management diagnosed from 1994–2015. | Biochemical remission in 80% (188 of 235) cases after a CT-based treatment decision vs. 93% (491 of 526) after an AVS-based treatment decision ($p < 0.001$). |
| Rossi GP [28] | 1311 PA patients. | Imaging did not detect the culprit adrenal in 28% of the surgically cured unilateral PA patients. The clinical outcome did not differ significantly between the imaging-positive and imaging-negative patients. |
| Surgery is the suggested treatment for monolateral PA | | |
| Satoh M [34] | 326 PA patients who had received MRA treatment (n = 152) or adrenalectomy (n = 174). | Clinical outcomes were not different in after MRAs or adrenalectomy, except for a reduction in the number of antihypertensive drugs after surgery ($p < 0.001$). |
| Wu V-C [35] | 858 unilateral PA cases among 1220 PA patients and 1210 essential hypertension controls. | Adrenalectomy was associated with lower all-cause mortality of unilateral PA patients, compared to controls ($p = 0.017$). More beneficial effect of adrenalectomy over MRA treatment on long-term MACE ($p < 0.001$), atrial fibrillation ($p < 0.001$), and congestive heart failure ($p < 0.001$) in unilateral PA patients. |
| Rossi G. P [52] | 1125 consecutively newly diagnosed hypertensive patients (PA, PH, and IHA). | The medical treatment of PA patients was associated with an increase of 82% of relative risk of atrial fibrillation compared with APA (treated with adrenalectomy) and PH ($p = 0.025$). |

**Table 5.** *Cont.*

| First Author | Cohort Described | Significant Findings |
|---|---|---|
| MRA is able to reduce cardiovascular risk in patients with PA | | |
| Catena C [49] | 54 consecutive patients who received a diagnosis of PA between 1994 and 2001. | Cardiovascular outcome (myocardial infarction, stroke, any type of revascularization procedure, and arrhythmias) was similar to patients with PA treated with adrenalectomy vs. MRAs ($p = 0.71$). |
| Interpretation of AVS in selected cases (inadequate catheterization, contrast allergy) | | |
| Younes N [41] | 7 patients with previous allergic reactions to ICM were prepared for AVS with 3 doses of 7.5 mg dexamethasone. | Despite adequate serum cortisol suppression following dexamethasone, the basal and post-ACTH selectivity index confirmed adequate cannulation of both adrenal veins. No allergic reactions were reported. |
| Acharya R [42] | Retrospective review of 8 patients with bilateral adrenal masses and AICS (AVS 2008–2016 for cortisol and epinephrine with dexamethasone suppression). | AVS was useful in excluding unilateral adenoma as the source of AICS among patients with bilateral adrenal masses and AICS. |
| Ceolotto G [43] | 136 patients with biochemically confirmed PA, who wished to pursue the surgical cure. | Biochemical cure after adrenalectomy was used to assess the accuracy of LI calculated by using androstenedione, metanephrine and normetanephrine compared to cortisol. The accuracy of LI calculated with the different biomarkers was high for all biomarkers and showed no significant differences ($p < 0.0001$). |
| Christou F [44] | 125 PA patients. | Assessment of SIs of cortisol, free metanephrine, and the FTMR indices for the AVS procedure. Confirmation that free metanephrine-based SIs are better than those based on cortisol. |

Abbreviations: AICS: ACTH-independent Cushing's syndrome, PA: primary aldosteronism, PH: primary hypertension, IHA: idiopathic hyperaldosteronism, MRA: mineralocorticoid receptor antagonist, MACE: major cardiovascular events, CT: computed tomography, AVS: adrenal venous sampling, ICM: iodine contrast media, LI: lateralization index, SI: selectivity index, FTMR: free-to-total metanephrine ratio.

**Author Contributions:** Conceptualization, methodology, and writing—original draft preparation, I.T., C.M. and F.C.; supervision, writing—review and editing, C.S. (Chiara Sabbadin), C.S. (Carla Scaroni) and F.C. All authors have read and agreed to the published version of the manuscript.

**Funding:** This research received no external funding.

**Institutional Review Board Statement:** Not applicable in a review.

**Informed Consent Statement:** Written informed consent was obtained from the patients to publish this paper.

**Data Availability Statement:** Not applicable.

**Conflicts of Interest:** The authors declare no conflict of interest.

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
