# Peer review of "Imaging or Adrenal Vein Sampling Approach in Primary Aldosteronism? A Patient-Based Approach"

_tomography, doi:10.3390/tomography8060228_

Round 1

Reviewer 1 Report

This is an interesting review evaluating the possible alternative tools for PA subtyping, also providing a clinical point of view by describing different cases and the diagnostic and therapeutic approaches used in each patient.

I would give full consideration to this paper. Only few points need to be addressed:

-     - Although the paper is generally well written, I would prefer a more formal language, by avoiding contractions or abbreviation (e.g. “it’s made”, line 89; “can’t”, line 341) and the repetition of the same verbs or expressions.

-     - In Case 3 the authors discussed the alternative options to cortisol in determining the selectivity and lateralization indexes. Metanephrine concentrations, along with catecholamines, are used by some groups to define the correct incannulation and lateralization. I think this could an interesting point to integrate and expand.

-  - Figure 1 is not clear. Is it supposed to be a flow chart or simply a figurative representation of the diagnostic options? Please revise it.

-  -  Line 204 should be “pmol/L”.

-    -  Layout of Table 2 and Table 4 should be revised.

Author Response

Reviewer 1

This is an interesting review evaluating the possible alternative tools for PA subtyping, also providing a clinical point of view by describing different cases and the diagnostic and therapeutic approaches used in each patient.

[Reply to reviewer #1]: We earnestly appreciate Editors/Reviewers’ warm work and hope that the correction will meet their final approval. We tried our best to improve the manuscript and made some changes in the manuscript. These changes will not influence the content and framework of the paper. We look forward to hearing from you regarding our submission. We would be glad to respond to any further questions and comments that you may have. Once again, thank you very much for your comments and suggestions.

I would give full consideration to this paper. Only few points need to be addressed:

  • Although the paper is generally well written, I would prefer a more formal language, by avoiding contractions or abbreviation (e.g. “it’s made”, line 89; “can’t”, line 341) and the repetition of the same verbs or expressions.

[Reply to reviewer #1]: thank so much, an English speaker reviewed the paper.

  • In Case 3 the authors discussed the alternative options to cortisol in determining the selectivity and lateralization indexes. Metanephrine concentrations, along with catecholamines, are used by some groups to define the correct incannulation and lateralization. I think this could an interesting point to integrate and expand.

[Reply to reviewer #1]: we agree and discussed the proposed item, adding also some literature

  • Figure 1 is not clear. Is it supposed to be a flow chart or simply a figurative representation of the diagnostic options? Please revise it.

[Reply to reviewer #1]: it is not a flow chart, it is a figure that try to “balance” the diagnostic options, pros and cons of the AVS in patients with PA. according to your suggestion, we add it in the description of the figure.

  • Line 204 should be “pmol/L”.

[Reply to reviewer #1]: we apologize for the mistake, we fix it.

  • Layout of Table 2 and Table 4 should be revised.

[Reply to reviewer #1]: we have revised all tables in order to enhance their clear and smart view, according to journal’s template.

Reviewer 2 Report

Please find my comments below and improve your paper:

1. line 39: national/international society 

2. The review addresses an interesting topic, but the methodology used is not transparent, and the results/conclusions suffer as a result of this problem. 

3. the lack of clearly-formulated objectives, which makes the entire construction of the review difficult to understand

4. no specified methodology for the collection of the reviewed data

5. the conclusions should respond to the previously formulated objectives, and compare the results with similar reviews existing in the literature; also, a section dedicated to the limitations of the present review is required.

6. a general review of the grammar and syntax should be conducted because there are many phrases where the articles are missing, or several expressions are improperly used.

7. synthesize the results in a graphical way (tables, figures), in order to add more consistency and comprehensibility to the review.

8. data should be summarized in the tables after each aspects of the work to summarize it.

9. please consider to cite these works:

  • DOI: 10.5114/ada.2021.107926

  • DOI: 10.5114/ada.2020.94181

Author Response

Reveiwer 2

Please find my comments below and improve your paper:

[Reply to reviewer #2]: We appreciate for Editors/Reviewers’ warm work earnestly, and hope that the correction will meet with approval. We tried our best to improve the manuscript and made some changes in the manuscript. These changes will not influence the content and framework of the paper. We look forward to hearing from you regarding our submission. We would be glad to respond to any further questions and comments that you may have. Once again, thank you very much for your comments and suggestions.

  1. line 39: national/international society

[Reply to reviewer #2]: we add that Endocrine Society is the international (leading) society in the field of Endocrinology.

  1. The review addresses an interesting topic, but the methodology used is not transparent, and the results/conclusions suffer as a result of this problem.

[Reply to reviewer #2]: thank so much for the suggestion. We write a narrative review, based on a patient-based personalized approach. We add all these considerations in the manuscript, and in the conclusions of the text.

  1. the lack of clearly-formulated objectives, which makes the entire construction of the review difficult to understand

[Reply to reviewer #2]: we added that our objective is to discuss the clinical applications of AVS.

  1. no specified methodology for the collection of the reviewed data

[Reply to reviewer #2]: we have specified that it is a narrative review, therefore no systematic review of literature has been performed.

  1. the conclusions should respond to the previously formulated objectives, and compare the results with similar reviews existing in the literature; also, a section dedicated to the limitations of the present review is required.

[Reply to reviewer #2]: we added the conclusions required and some sentences regarding the limitations of the study.

  1. a general review of the grammar and syntax should be conducted because there are many phrases where the articles are missing, or several expressions are improperly used.

[Reply to reviewer #2]: according also to reviewer 1, an English speaker reviewed the paper.

  1. synthesize the results in a graphical way (tables, figures), in order to add more consistency and comprehensibility to the review.

[Reply to reviewer #2]: we have modified figure 1 and all tables in order to add the information required.

  1. data should be summarized in the tables after each aspects of the work to summarize it.

[Reply to reviewer #2]: see previous reply, and only table 1 is the sum of all patients.

  1. please consider to cite these works: DOI: 10.5114/ada.2021.107926, DOI: 10.5114/ada.2020.94181

[Reply to reviewer #2]: thank so much for the suggestions. Nonetheless, with all due respect to the reviewer, we felt that the suggested points are not completely correct. The referred titles for DOI: 10.5114/ada.2021.107926 is “The influence of TNF-α on the expression profile of key enzymes of steroidogenesis in H295R cells” and for DOI: 10.5114/ada.2020.94181 is “Analysis of the influence of adalimumab to the expression pattern of mRNA and protein of TGF-β1-3 in dermal fibroblast exposed to lipopolysaccharide”.

The first article evaluates the transcriptional activity of some enzyme of the steroidogenesis in cell line (not human) after TNFa, among them aldosterone synthase expression. The second one analyses the expression profile of TGF-β exposed to inflammation or adalimumab (anti-TNF drug).

Therefore, I am not able to cite such articles in a paper that describe clinically the role of vein sampling in patients with primary aldosteronism.

Round 2

Reviewer 2 Report

Even if this is a narrative review, the authors had to use any key for searching articles. It has to be clearly described. Each sub-section should be summarized in the table, including the number of references and last name of the first authors, type of paper, number of cases, significant findings, etc.

Author Response

Reviewer 2

Even if this is a narrative review, the authors had to use any key for searching articles. It has to be clearly described. Each sub-section should be summarized in the table, including the number of references and last name of the first authors, type of paper, number of cases, significant findings, etc.

[Reply to reviewer #2]: We earnestly appreciate Reviewer’s work and hope that the correction will meet the final approval. We tried our best to improve the manuscript and made the required changes in the manuscript.

We look forward to hearing from you regarding our submission. We would be glad to respond to any further questions and comments that you may have. Once again, thank you very much for your comments and suggestions.

We added:

  • The keywords used for searching articles.
  • We added a final table, as required, that indicates the best references and number of cases reported.